# The mediating role of knowledge sharing behavior and the moderating role of digital mindset: Evidence in Vietnam

Tu Ngoc Tran ◉*, Nguyen Thi Thanh Tam, Khai Minh Tu

Faculty of Business Administration, Saigon University, Ho Chi Minh City, Vietnam

* tntu@sgu.edu.vn

## Abstract

This study examined the mediating roles of knowledge sharing in the effect of organizational-level factors on knowledge application in Vietnamese commercial banks and the moderating role of the digital mindset. Through the application of Partial Least Squares Structural Equation Modeling (PLS-SEM), this study reveals that knowledge sharing serves as a crucial mediating mechanism linking organizational-level factors, namely, reward system, organizational commitment, organizational strategy, and organizational structure, to knowledge application. Additionally, digital mindsets directly affect knowledge applications and strengthen the impact of knowledge sharing on knowledge applications. The findings of this study contribute to consolidating fundamental theories such as social exchange theory, socio-technical system theory, and dynamic capability theory and provide important practical implications for building human resource management policies and promoting digital transformation at commercial banks in Vietnam.

## Introduction

The explosion of the Fourth Industrial Revolution with advanced technologies such as artificial intelligence (AI), big data, the Internet of Things (IoT), and cloud computing has fundamentally changed the way businesses create value, make decisions, and maintain competitive advantage. In this context, knowledge, especially specialized and innovative knowledge, is a supporting factor and a central resource driving productivity and operational efficiency [1,2]. Globally, pioneering organizations are promoting the construction of knowledge management systems to maximize internal capacity, in which knowledge sharing (KS) is increasingly considered a decisive factor for adaptability and continuous innovation [3]. In Vietnam, many businesses, especially in the technology, finance, and service sectors, have begun to focus on exploiting knowledge as an intangible asset to adapt to the increasingly competitive and digital business environment.

**Data availability statement:** The dataset generated and analyzed during this study is available in the Dryad data repository at: DOI: 10.5061/dryad.02v6wwqhh.

**Funding:** The Impact of Organizational Factors on Knowledge Application Behavior: The Mediating Role of Knowledge Sharing and the Moderating Role of Digital Mindset); Project code: CSA.2025.056, is funded by Saigon University.

**Competing interests:** The authors have declared that no competing interests exist.

In this context, knowledge has become a core strategic asset that plays a decisive role in promoting innovation, enhancing competitiveness, and ensuring the sustainable development of enterprises. In the knowledge management process chain, KS is considered a key link, allowing the transformation of tacit knowledge into explicit knowledge, thereby expanding the ability to access and apply knowledge throughout the organization [4].

Empirical studies have confirmed the mediating role of KS in the relationship between organizational factors and innovation [5,6]. KS spreads specialized information and promotes mutual learning, contributing to the integration of ideas to form effective and innovative solutions [7,8]. However, KS is not sufficient to create practical value if it is not transformed into an applied action. Therefore, knowledge application is considered a decisive stage in the knowledge value chain when shared knowledge is integrated and applied to solve problems, innovate processes, or make effective decisions [9,10].

In addition to organizational factors such as reward systems, organizational commitment, strategy, and organizational structure [11,12,13], a personal factor that is of particular interest to researchers in the context of digital transformation is the digital mindset (DM). According to [14], DM is an individual's cognitive tendency toward readiness to respond, apply, and exploit digital technologies at work. DM manifests as dynamic capabilities that help individuals quickly integrate, adjust, and apply knowledge in a digital environment [15,16].

According to the Sociotechnical Systems theory, the effectiveness of knowledge application (KA) depends on the interaction between human and technological systems [17]. In this context, digital thinking helps individuals interact more effectively with technological systems, thereby increasing the effectiveness of KS and applications [18]. Although many studies worldwide have mentioned the mediating role of KS in the relationship between organizational factors and innovation and emphasized the contribution of digital thinking in the context of digital transformation [19,14], most of these studies mainly use data from high-tech enterprise environments or developed regions. In Vietnam, although the concept of KS and application has been mentioned in several studies, the integration of digital thinking as a moderating variable in the relationship between the two variables has not been considered significant, especially in the commercial banking sector, which is undergoing many technological mergers and rapid transformations.

Therefore, this study was conducted to clarify the mediating role of KS in the relationship between organizational factors and knowledge application, and to explore the moderating role of digital mindset on the relationship between KS and KA in the context of rapid digital transformation in Vietnamese commercial banks.

## Literature review, previous research and hypothesis development

### Literature review

This section provides an overview of the main theoretical foundations and previous studies that inform the research model. The discussion begins with key theories that explain knowledge sharing behavior, including Social Exchange Theory, the Theory

of Planned Behavior, and Social Cognitive Theory. It then incorporates Dynamic Capabilities Theory and Sociotechnical Systems Theory to highlight the role of digital mindset in knowledge management and application.

**Theories explaining individual behavior.** Social Exchange Theory (SET) suggests that social relationships are formed and maintained on the basis of exchanging benefits, as indicated by Homans [20]; Blau [21]. Accordingly, social behavior only occurs when the benefits received are considered greater than the costs incurred. In organizational settings, SET is often used to explain the motivation for knowledge sharing, which is considered a trade-off between costs and benefits [22]. When the work environment creates favorable conditions such as trust among colleagues, support from superiors, or a fair organizational culture, employees tend to actively share knowledge – a voluntary behavior that depends on subjective perceptions of exchange value [23,24].

SET emphasizes external interactions and social situations that facilitate knowledge sharing and exchange. At the same time, Social Cognitive Theory (SCT) provides an additional explanation in terms of internal psychological mechanisms. According to this theory, human behavior is the result of interactions among personal factors (beliefs and skills), behavior, and environment [25]. Regarding knowledge-sharing behavior, SCT emphasizes the importance of self-efficacy in promoting proactive behavior. It means that employees who believe they are capable of sharing knowledge effectively will tend to be more proactive in communicating information, supporting colleagues, and promoting innovation at the organizational level [26,27].

Besides SET and SCT, the Theory of Planned Behavior (TPB) emphasizes the role of behavioral intention in forming knowledge-sharing behavior. According to [28], attitudes, subjective norms, and perceived behavioral control are key factors that determine an individual's intention, which in turn influences their behavior. In the organizational context, organizational commitment reflects a positive attitude toward knowledge sharing, while perceived competence in knowledge sharing reflects perceived behavioral control [29]. Combining the three theories helps to explain better how knowledge sharing behavior stems from personal motivation and beliefs, and is influenced by the social environment, thereby promoting innovation in organizations.

**Theories of dynamic capabilities and socio-technical systems.** In the era of substantial digital transformation, digital mindset – the way each individual shapes their perception and readiness to embrace technology – is increasingly seen as a decisive factor, helping organizations convert knowledge into practical innovation activities [19,16]. This mindset enables individuals to adapt swiftly to technological changes, leverage digital tools, and creatively apply knowledge to drive innovation.

According to the Dynamic Capabilities Theory of [15], dynamic capabilities are understood as the ability of a business or individual to integrate, build, and restructure internal and external resources, including knowledge, to adapt to an ever-changing environment. In this framework, digital thinking is viewed as a manifestation of dynamic capabilities at the individual level, demonstrating the ability to flexibly adapt to new technologies and quickly exploit and apply knowledge in a digital context [16].

Furthermore, according to Socio-Technical Systems Theory, an organization consists of the social part (people and structure) and the technical part (technology and processes). When these two parts work together, knowledge sharing and innovation will occur more effectively [17,30]. In this context, a digital mindset refers to one's ability to identify, understand, and apply competently within a technology system to increase the effectiveness of KS and its application. People who have developed a digital mindset are more skilled in learning, technology use, and work practice innovation; therefore, they are more likely to adapt with less difficulty, especially in more automated and digitalized organizational contexts [19].

## Previous research

Numerous empirical studies have confirmed the essential role of KS and application in improving work efficiency. Specifically, Boateng & Agyemang [7] showed that KS and knowledge application (KA) both have a positive impact on service recovery performance (SRP), with KA having a more substantial impact. Besides, Sawana & Nurhattati [31] systematized

70 factors influencing knowledge-sharing behavior and pointed out the prominent role of organizational culture and transformational leadership based on theories such as social exchange, social cognition, and social capital. Another study by Kmieciak [6] emphasized trust (vertical and horizontal) as an antecedent to promoting KS, affecting innovation behavior. Other studies, such as Safdar et al. [27]; Shehab et al. [32], also confirmed the moderating role of self-efficacy in knowledge-sharing behavior. Meanwhile, Binsaeed et al. [5]; Yang et al. [33] pointed out that KS mediates the relationship between organizational factors such as network capabilities or digital transformation intentions and innovation outcomes or adaptive capacity. However, most studies still focus on each organizational, individual, or technological factor separately, while few studies integrate these factors in a comprehensive model, especially in the context of digital transformation in developing countries – this is the gap this study aims to fill.

Therefore, although many previous studies have confirmed the mediating role of knowledge-sharing in promoting organizational performance and innovation [5,7], as well as identified factors influencing knowledge-sharing behavior [27,31], most of them were conducted outside Vietnam and have not fully considered the context of digital transformation. In particular, the moderating role of the digital mindset – an important factor in the digital environment – has not been deeply explored in the relationship between KS and application. As a result, this study aims to fill the gap by simultaneously testing the mediating role of KS and the moderating role of the digital mindset, thereby contributing theoretically and practically to knowledge management in Vietnamese enterprises during the digital transformation period.

## Hypothesis development

**Reward system affects knowledge sharing.** In a rapidly developing knowledge economy, KS among employees is considered an essential factor for organizations to improve their innovation capabilities and maintain competitive advantages [11,34]. However, many individuals remain hesitant to share knowledge because of concerns about losing personal advantages or not receiving appropriate recognition [35]. In this context, reward systems, including both material and spiritual rewards, have been proposed as effective incentive tools to promote knowledge-sharing behavior in organizations [36,37].

The reward system includes policies that encourage and recognize employee contributions through extrinsic rewards (bonuses and promotions) and intrinsic rewards (recognition and job satisfaction) [38]. According to Lin & Lo [34], which is an exchange mechanism in social exchange theory in which employees only share when they feel that the benefits they gain exceed the costs they spend. Previous studies have also confirmed that transparent and fair reward systems enhance employees' motivation to share knowledge [11,36]. Conversely, poorly designed systems can hinder this behavior regardless of the supporting technology platform [37].

From the above arguments, the authors propose the following research hypothesis:

*Hypothesis 1 (H1): The reward system in the organization has a positive influence on employees' knowledge sharing behavior in Vietnamese commercial banks.*

**Organizational commitment influences knowledge sharing.** Among the organizational-level factors, commitment has been widely recognized as a crucial determinant of employees' willingness to share knowledge. In this context, organizational commitment is the extent to which employees are emotionally attached, willing to participate in the organization's activities, and make every effort to achieve a common goal [39]. Affective, continuance, and normative commitment are the three main components of organizational commitment [40,41,42]. In the modern working environment, organizational commitment plays an important role in retaining talent and building an internal knowledge-sharing culture.

Previous studies have demonstrated that organizational commitment is positively associated with knowledge-sharing behaviors [40,13]. When employees feel emotionally attached to and trust their organization, they are more willing to

share knowledge and experience to support collective growth. Demirel and Goc [43] argue that the (affective) commitment to the organization significantly transfers information from one member to another. In the same arguments, the recent work of [40] confirms that organizational commitment is an adequate predictor of knowledge-sharing behavior, finally leading to improved organizational performance.

Consequencely, we can propose the hypothesis as follows:

*Hypothesis 2 (H2): Organizational commitment has a positive influence on employees' knowledge sharing behavior in Vietnamese commercial banks.*

**Organizational strategy influences knowledge sharing.** According to Grant [44], organizational strategy refers to how a business integrates and exploits knowledge to create core competencies and competitive advantages. In this context, "operational excellence" can be understood as a value proposition strategy that emphasizes improving the efficiency and reliability of processes, in order to deliver products at competitive prices [45]. This approach demonstrates that organizational strategy not only guides business operations but also directly influences the way knowledge is shared, leveraged, and utilized within the enterprise. In the knowledge economy, this strategy is a plan of action and foundation for promoting innovation, improving performance, and developing resources through effective knowledge management and sharing [46,12].

The relationship between organizational strategy and knowledge-sharing behaviors has received increasing attention. A strategy focusing on innovation, learning, and collaboration facilitates a culture of KS [47]. Organizational strategy also governs factors such as structure, reward policies, and technology, which affect knowledge-sharing behavior [11]. At the same time, the study of Razak et al. [12] emphasized that organizational strategy needs to be synchronized with a sharing culture, especially commitment and internal support systems.

*Hypothesis 3 (H3): Organizational strategy has a positive influence on employees' knowledge sharing behavior in Vietnamese commercial banks.*

**Organizational structure affects knowledge sharing.** According to Mahmoudsalehi et al. [48], an organizational structure includes power centralization, formalization, specialization, functional integration, and flexibility. These factors shape how information and knowledge are created, shared, and used within an organization [49]. Organizational structure not only affects management effectiveness but also knowledge-sharing behavior by regulating the ability to connect and coordinate among members of the organization [50].

Empirical studies have shown that a decentralized, less normative, and highly integrated structure promotes social interaction, enhances internal communication, and encourages knowledge-sharing behaviors between individuals and departments [51,49,52]. On the contrary, a structure that is too rigid, highly centralized, and has many administrative procedures can reduce initiative and creativity and create barriers to KS. In modern organizations, especially high-tech enterprises, building a flexible organizational structure and effective connections between functional units are key factors in promoting knowledge management. Based on these theoretical and empirical bases, we propose the following hypotheses:

*Hypothesis 4 (H4): Organizational structure has a positive influence on employees' knowledge sharing behavior in Vietnamese commercial banks.*

**Knowledge sharing affects knowledge application.** Knowledge application (KA) refers to exploiting and applying existing knowledge in an organization to solve problems, improve processes, develop products, and make effective decisions [7,9,53]. This step determines the practical value of knowledge management when knowledge is stored,

shared, and integrated into the organization's operations to create an efficient and sustainable competitive advantage [10]. Similarly, KA is also the process of applying knowledge into practice to guide strategy, solve problems and improve operational efficiency [54].

Several studies have shown that KS is an important factor [C. 55] and a prerequisite for enhancing the effectiveness of knowledge applications. Choi et al. [8] demonstrated that KS among team members is positively associated with the ability to apply knowledge and team performance. Similarly, the findings of Boateng & Agyemang [7]; Iqbal et al. [56] also maintain that knowledge workers transfer and share experiences, skills, and knowledge; this knowledge can be translated into innovation and process improvement, which in turn helps organizations to perform better.

*Hypothesis 5 (H5): Knowledge sharing positively affects knowledge applicationin in Vietnamese commercial banks.*

**The moderating role of Digital Mindset (DM).**

a)  Digital Mindset moderates the effect of knowledge sharing on knowledge application

In the context of substantial digital transformation, DM – the mindset of being ready to adapt and exploit digital technology – is considered a key individual factor that promotes the effectiveness of transforming knowledge into innovative actions in organizations [57,14,16]. Although KS is a prerequisite for knowledge to be applied in practice, this ability does not occur automatically. However, it depends mainly on the attitude and technological capacity of the knowledge sharer and receiver [4,3].

According to Dynamic Capabilities Theory [15], individuals with high digital mindsets are able to integrate, restructure, and apply knowledge flexibly in a rapidly changing technological environment. Therefore, the Digital Mindset can act as a positive moderator, strengthening the relationship between KS and knowledge application, helping employees share information, and proactively transforming knowledge into specific improvements or actions [19,58].

In addition, according to Sociotechnical Systems Theory, the effectiveness of knowledge application depends on the level of interaction between the human and technological systems [17]. Individuals with developed digital thinking often interact more effectively with technologies, thereby facilitating the application of knowledge to work [58,18]. This means that with high digital thinking, the impact of KS on knowledge applications will become stronger.

Based on the above arguments, we propose the following hypothesis:

*Hypothesis 6 (H6): Digital Mindset positively moderates the effect of knowledge sharing on knowledge application in Vietnamese commercial banks.*

b)  Digital Mindset moderates the effect of Organizational commitment on knowledge sharing

In the modern workplace, organizational commitment (OC) is considered a key factor that promotes KS behavior among employees, as analysed before. OC creates a psychological and volitional attachment between individuals and the organization, increasing the tendency to share information, experience, and capabilities to support common goals [59]. However, a digital mindset can significantly influence this relationship. Specifically, although employees have a high level of commitment to the organization, if they lack digital mindset, that is, lack of openness, adaptability, and willingness to adopt digital technologies, the ability to share knowledge through digital platforms will be limited [60]. Mindset acts as a moderator, helping to maximize the positive impact of organizational commitment on KS behavior, especially in organizations undergoing substantial digital transformation [61,62].

*Hypothesis 7 (H7): Digital Mindset positively moderates the effect of Organizational commitment on Knowledge sharing in Vietnamese commercial banks.*

c) Digital Mindset moderates the effect of Organizational strategy on knowledge sharing

The effectiveness of organizational strategy in promoting KS behavior depends on the rationality of planning and the extent to which employees realize that strategy. In Vietnamese enterprises in digital transformation, many strategies are well-developed but face difficulties in practical implementation [63]. The lack of synchronization between strategic planning and implementation may stem from a lack of digital capabilities, innovative thinking, or consensus from the human resources – those who directly turn strategies into actions [64,65]. In this context, Digital Mindset plays an important role as a moderator, helping to enhance the impact of organizational strategy on KS behavior. Employees with a high digital mindset are often better able to connect with technology tools, thereby proactively sharing knowledge and supporting each other through digital platforms [66]

When employees have a high level of digital mindset, organizational strategies can be easily translated into concrete behaviors by applying knowledge management systems, internal social networks, and digital collaboration platforms [58,67]. Digital mindset helps employees develop digital capabilities, enhance adaptability, and actively leverage digital tools to achieve strategic goals [19]. Conversely, if employees lack the necessary capabilities and hold negative attitudes toward technology, even well-defined strategies may stagnate or become counterproductive due to insufficient engagement and ineffective execution in the digital environment [67,62].

This shows that digital thinking plays a clear moderating role, changing the level of influence of organizational strategy on KS within the organization [62]. According to [68], digital transformation only achieves the desired results when there is a synchronization between employees' digital capabilities and the enterprise's strategic orientation.

*Hypothesis 8 (H8): Digital Mindset positively moderates the effect of Organizational strategy on Knowledge sharing in Vietnamese commercial banks.*

## Research model and research method

### Research model

The research model (Fig 1) examines the impact of organizational factors, including reward system (RS), organizational commitment (OC), organizational strategy (OS), and organizational structure (OT), on knowledge sharing (KS). Furthermore, KS is expected to positively influence knowledge application (KA). In addition, digital mindset (DM) is hypothesized

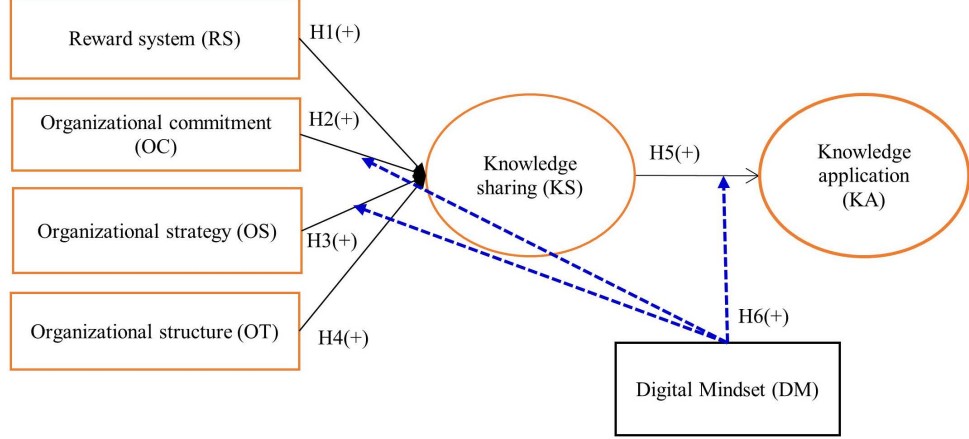

**Fig 1. Research model.**

to play a moderating role, strengthening the effects of organizational factors on KS as well as the relationship between KS and knowledge application.

## Research method

The research employed quantitative methods using partial least squares structural equation Modeling (PLS-SEM). Data analysis was conducted using SmartPLS, including descriptive statistics, validation of the observed constructs through factor loadings, assessment of convergent validity and composite reliability, evaluation of discriminant validity within the measurement model, and checks for multicollinearity.

The degree to which the independent variables explain the variation in the dependent variable was tested based on the $R^2$ and adjusted $R^2$ indices. In addition, the quantitative effect of each predictor variable on the dependent variable was measured using the f-squared ($f^2$) index. The effect sizes based on $f^2$ were categorized into three levels. Besides, as suggested by Cohen [69], the rule for $f^2$ as follows: small effect if $f^2 \geq 0.02$, medium effect if $f^2 \geq 0.15$, and large effect if $f^2 \geq 0.35$. These thresholds help assess the statistical and practical significance of relationships within the research model.

## Sample size

Data collection techniques involved developing structured questionnaires and applying simple random sampling to ensure representation across various branches of commercial banks in Ho Chi Minh City, Vietnam. The questionnaire was designed to assess the key constructs of the research model, with multiple items allocated to each construct. Responses were measured using a 5-point Likert scale, allowing participants to indicate their level of agreement with statements ranging from "completely disagree" (1) to "completely agree" (5). This approach provided quantifiable data suitable for subsequent statistical analyses.

The survey was conducted with experts, directors, and employees working at branches of commercial banks in Ho Chi Minh City, Vietnam, from February to June 2025. A total of 350 questionnaires were distributed via email and QR codes, and after excluding eight incomplete responses, 324 valid questionnaires were retained for the analysis, yielding a high response rate of 92.6%. Prior to accessing the questionnaire, participants were provided with an information sheet detailing the study's objectives, procedures, potential risks and benefits, and data protection measures. Online informed consent was obtained by requiring participants to click "I agree to participate" before proceeding with the survey, and they were free to withdraw at any time. Only respondents who explicitly confirmed consent were able to complete the questionnaire. This procedure, including the use of online informed consent, was reviewed and approved by the Institutional Review Board/Ethics Committee of Saigon University, ensuring compliance with the Declaration of Helsinki.

## Research results and discussions

### Research results

This section presents the empirical results obtained from the PLS-SEM analysis. The findings include assessments of reliability and validity, discriminant validity, explanatory power of the model, effect sizes, and hypothesis testing, supported by tables and figures.

From Table 1, Cronbach's Alpha, Composite Reliability and AVE indices all exceeded the thresholds of 0.7 and 0.5, indicating that the scales had high reliability and good convergent validity, ensuring suitability for further analysis in the research model. Discriminant validity assessment using the heterotrait-monotrait (HTMT) ratio is presented in Table 2.

Table 2 shows that all values were less than 0.85, indicating that the variables in the model achieved good discriminant validity. This confirms that the measurement constructs were separate and did not overlap in the research model.

Fig 2 was generated using the bootstrapping method with 5,000 resample iterations conducted using SmartPLS software. Bootstrapping is used to evaluate the reliability and stability of the estimated path coefficients. This method enables the calculation of standard errors and t-values, which are critical factors in hypothesis testing.

**Table 1. Composite reliability, Cronbach's alpha, and AVE indices of all study constructs.**

|  | Cronbach's alpha | Composite reliability (rho_a) | Composite reliability (rho_c) | Average variance extracted (AVE) |
|---|---|---|---|---|
| DM | 0.997 | 0.997 | 0.998 | 0.995 |
| KA | 0.953 | 0.958 | 0.969 | 0.914 |
| KS | 0.989 | 0.989 | 0.992 | 0.978 |
| OC | 0.932 | 0.934 | 0.957 | 0.881 |
| OS | 0.913 | 0.914 | 0.946 | 0.853 |
| OT | 0.983 | 0.984 | 0.989 | 0.967 |
| RS | 0.985 | 0.987 | 0.99 | 0.971 |

**Table 2. Heterotrait-monotrait ratio (HTMT).**

|  | DM | KA | KS | OC | OS | OT | RS | DM x OC | DM x OS | DM x KS |
|---|---|---|---|---|---|---|---|---|---|---|
| DM |  |  |  |  |  |  |  |  |  |  |
| KA | 0.677 |  |  |  |  |  |  |  |  |  |
| KS | 0.626 | 0.700 |  |  |  |  |  |  |  |  |
| OC | 0.308 | 0.368 | 0.462 |  |  |  |  |  |  |  |
| OS | 0.407 | 0.378 | 0.496 | 0.410 |  |  |  |  |  |  |
| OT | 0.372 | 0.617 | 0.634 | 0.296 | 0.361 |  |  |  |  |  |
| RS | 0.424 | 0.497 | 0.667 | 0.493 | 0.470 | 0.572 |  |  |  |  |
| DM x OC | 0.161 | 0.057 | 0.096 | 0.299 | 0.082 | 0.041 | 0.081 |  |  |  |
| DM x OS | 0.205 | 0.039 | 0.023 | 0.088 | 0.190 | 0.154 | 0.091 | 0.307 |  |  |
| DM x KS | 0.356 | 0.193 | 0.463 | 0.088 | 0.021 | 0.191 | 0.165 | 0.075 | 0.304 |  |

The findings in Table 3 show that the R-square value of the KS variable reached 0.694, indicating that 69.4% of the variation in KS was explained by organizational factors, including the reward system, organizational commitment, strategy, and organizational structure, reflecting a high level of explanation. Meanwhile, the R-square of the Knowledge Application (KA) variable reached 0.587, indicating that the model could explain 58.7% of the variation in KA through KS, a digital mindset, and the interaction between these two factors.

The following sections show the results of VIF values and f-square (Table 4). The details of analysis are as follows:

First, Table 4 shows that all the VIF values were lower than the threshold of 5. This showed no signs of multicollinearity among the independent variables in the model. Thus, the linear relationships between the variables are guaranteed to be reliable and unaffected by the interdependence of the explanatory variables.

Second, we define the effect size based on the f-square ratio. Based on the VIF and f-square tables, all relationships in the model have f-square ($f^2$) values greater than 0, indicating that each independent variable contributes to some extent to the $R^2$ value of the dependent variable. Specifically, the relationships between DM and KS ($f^2 = 0.348$) and KS ◊ KA ($f^2 = 0.346$) are assessed to have medium effects and play an important role in explaining KS and KA. Similarly, DM ◊ KA ($f^2 = 0.253$) and OT ◊ KS ($f^2 = 0.186$) also shows medium effects, reflecting an apparent influence in the model. Meanwhile, the paths from OC to KS ($f^2 = 0.059$), OS to KS ($f^2 = 0.023$), RS to KS ($f^2 = 0.069$), DM × OC to KS ($f^2 = 0.057$), DM × OS to KS ($f^2 = 0.031$), and DM × KS to KA ($f^2 = 0.070$) only achieve small effects, indicating that their contributions are present but not strong. Overall, the $f^2$ results confirm that the model has good explanatory power, with KS playing a central role in shaping KA, while DM also shows a consistent impact on both KS and KA.

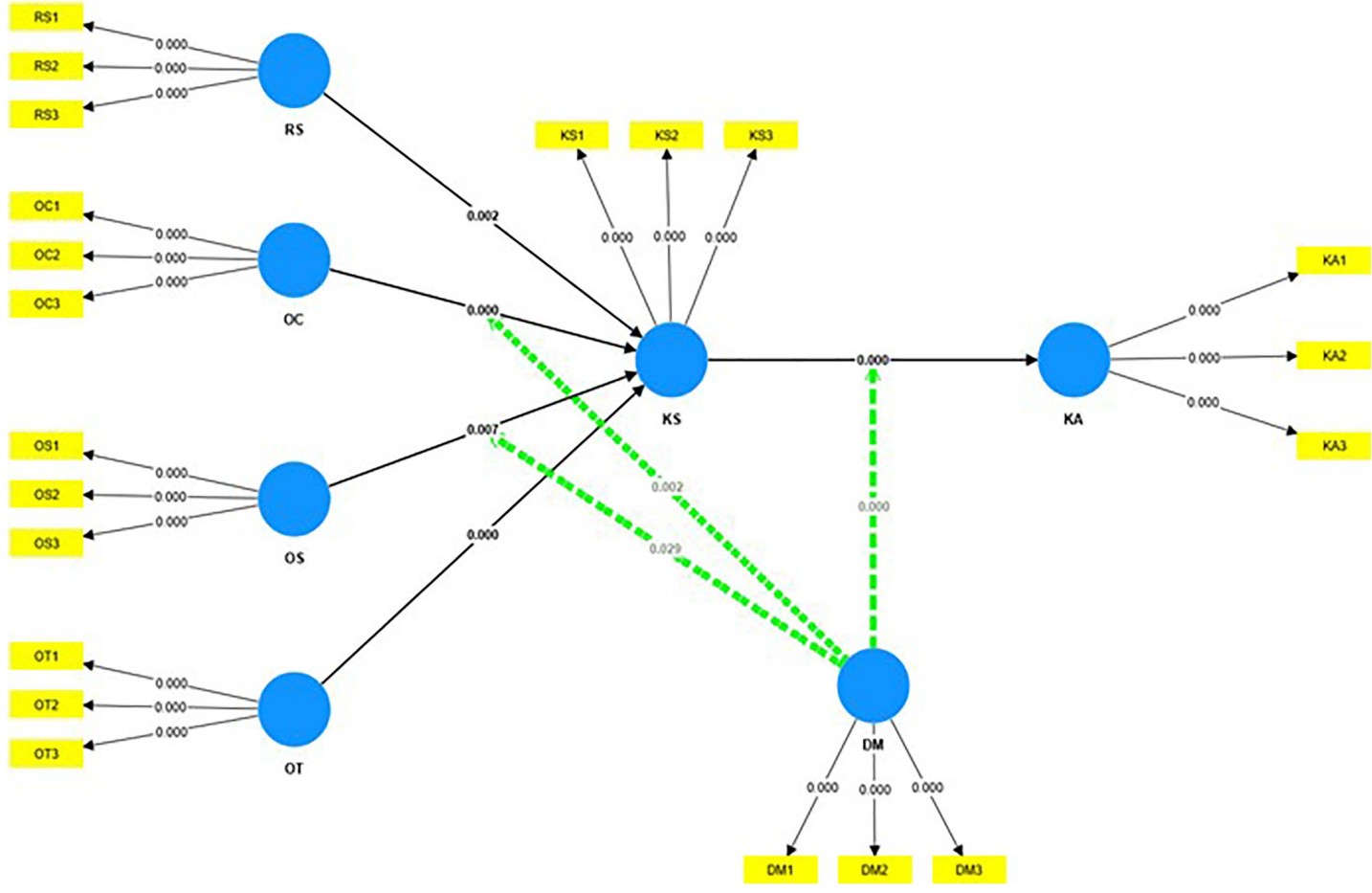

**Fig 2. Partial Least Squares Structural Equation Modeling (PLS-SEM) Model results.**

**Table 3. R-square and R-square adjusted.**

|  | R-square | R-square adjusted |
|---|---|---|
| KA | 0.587 | 0.583 |
| KS | 0.694 | 0.687 |

## Discussions

This section discusses the key findings of the study in relation to existing theories and prior empirical research. The analysis highlights how organizational factors and digital mindset influence knowledge sharing and application, and it evaluates the consistency of the results with previous studies.

The regression results in Table 5 show that all hypothesized relationships are statistically significant ($p < 0.05$), with positive path coefficients, indicating positive effects among the variables. The following analysis discusses these significant relationships and evaluates whether the findings are consistent with empirical studies or supported by relevant theories.

First, Digital Mindset (DM) has a statistically significant positive effect on KA because the coefficient is 0.415 and its p-value is 0.000 (less than 5%). This finding suggests that digital thinking plays an important role in translating knowledge

**Table 4. VIF and f square (f²).**

|  | VIF | f-square | Effect size |
|---|---|---|---|
| DM -> KA | 1.647 | 0.253 | Medium effect |
| DM -> KS | 1.387 | 0.348 | Medium effect |
| KS -> KA | 1.827 | 0.346 | Medium effect |
| OC -> KS | 1.551 | 0.059 | Small effect |
| OS -> KS | 1.408 | 0.023 | Small effect |
| OT -> KS | 1.538 | 0.186 | Medium effect |
| RS -> KS | 2.003 | 0.069 | Small effect |
| DM x OC -> KS | 1.333 | 0.057 | Small effect |
| DM x OS -> KS | 1.176 | 0.031 | Small effect |
| DM x KS -> KA | 1.281 | 0.070 | Small effect |

into actions that enhance the effectiveness of knowledge application in the firm. This finding is consistent with Dynamic Capabilities Theory [15], which argues that cognitive capacity and digital thinking restructuring are the foundation for organizations to adapt and innovate in the digital environment. At the same time, Sociotechnical Systems Theory [17] also emphasizes that the interaction between people and technology is a decisive factor in the successful application of knowledge. This research result is also similar to previous studies such as Solberg et al. [14]; Warner & Wäger [16], where the authors assert that DM promotes technological adaptation and is a fundamental factor for innovation and operational efficiency in organizations. The research results indicate that businesses need to focus on cultivating digital thinking for both leaders and employees to enhance technology adaptability, promote innovation and improve competitiveness. From a theoretical perspective, this finding highlights the role of digital thinking as a distinct form of dynamic capability during the digital transformation period and confirms that the connection between people and technology is the key factor in effectively exploiting and applying knowledge.

Second, the impact of KS on KA is strong and statistically significant (β = 0.511, p = 0.000). This confirms that KS is essential for bridging knowledge ownership and application actions at work. This finding is consistent with previous studies such as Boateng & Agyemang [7]; Choi et al. [8]; Iqbal et al. [56], who emphasized that KS is a prerequisite for improving the effectiveness of application and innovation in organizations. From a theoretical perspective, Social Cognitive Theory [25] also explains that individuals who have confidence in their ability to share knowledge will be more proactive in sharing behavior, thereby improving the effectiveness of knowledge application in work practice. This result confirms that KS has a positive impact on KA, meaning that the more effectively knowledge is shared, the easier it is for employees to transform knowledge into concrete actions. Knowledge sharing not only helps spread experience and skills within the organization but also creates a collaborative learning environment, encourages innovation, and supports faster problem-solving. Therefore, promoting knowledge sharing is a crucial strategy for improving the quality of knowledge application and enhancing the competitiveness of enterprises.

Organizational Commitment (OC) significantly affects KS (β = 0.167, p = 0.000), indicating that employees committed to the organization are willing to share knowledge to contribute to common development. This result is consistent with the studies of Imamoglu et al. [40]; Van Den Hooff & De Ridder [13] and is explained by Social Exchange Theory [21], in which knowledge-sharing behavior is a response to the benefits received from the organization. The findings suggest that high levels of employee commitment to the organization encourage knowledge sharing as a way of expressing commitment and gratitude. This commitment creates a positive cycle: more committed employees share more, and sharing in turn improves performance and strengthens their trust in the organization. Therefore, creating a fair work environment, recognizing contributions, and providing career development opportunities are key to enhancing organizational commitment and promoting knowledge sharing behavior.

**Table 5. Hypothesis testing obtained from SmartPLS4.**

| | Original sample | Sample mean | Standard deviation | T statistics | P values |
|---|---|---|---|---|---|
| DM -> KA | 0.415 | 0.413 | 0.072 | 5.726 | 0.000 |
| DM -> KS | 0.384 | 0.383 | 0.061 | 6.316 | 0.000 |
| KS -> KA | 0.511 | 0.510 | 0.072 | 7.123 | 0.000 |
| OC -> KS | 0.167 | 0.168 | 0.042 | 3.940 | 0.000 |
| OS -> KS | 0.100 | 0.098 | 0.037 | 2.712 | 0.007 |
| OT -> KS | 0.296 | 0.292 | 0.052 | 5.745 | 0.000 |
| RS -> KS | 0.206 | 0.206 | 0.068 | 3.027 | 0.002 |
| DM x OC -> KS | 0.138 | 0.138 | 0.044 | 3.111 | 0.002 |
| DM x OS -> KS | 0.100 | 0.099 | 0.046 | 2.179 | 0.029 |
| DM x KS -> KA | 0.153 | 0.151 | 0.041 | 3.716 | 0.000 |

Moreover, Organizational Strategy (OS) also positively impacts KS (β = 0.100, p = 0.007), suggesting that strategic orientations that emphasize innovation, learning, and internal collaboration will create a favorable foundation for KS. This result is similar to previous studies, such as Razak et al. [12]; Witherspoon et al. [47], in which strategic alignment with a sharing culture is a factor that promotes sharing behavior in organizations. From the results, it is suggested that organizational strategy plays a significant role in shaping knowledge sharing behavior, as shared development goals encourage employees to learn from and share experiences with one another. A strategy designed to support knowledge will help break down departmental barriers, create a collaborative culture, and improve innovation performance. Therefore, embedding KS into long-term vision and development strategy is an important factor in maintaining sustainable competitive advantage.

In addition, organizational structure (OT) positively affects KS (β = 0.296, p = 0.000), reflecting that organizations with flexible, less centralized structures will facilitate internal communication and KS. This finding is consistent with the results of Chen & Huang [51]; Flaszewska [49]; Islam et al. [52], which argue that de-bureaucracy and openness in organizational structures are the foundations for more effortless knowledge flow between individuals and departments. A flat and open organizational structure facilitates peer-to-peer collaboration, reduces administrative barriers, and promotes rapid and widespread knowledge sharing. Designing a flexible organization that promotes horizontal connections, and open communication is crucial to enhancing the effectiveness of knowledge sharing within an organization.

Finally, the impact of the Reward System (RS) on Knowledge Sharing behavior was positive and statistically significant (β = 0.384, p = 0.000), indicating that an effective reward mechanism acts as a key motivator for employees to proactively participate in the KS process in the organization. This result is consistent with that of Bartol & Srivastava [11]; Lin & Lo [34]; Opoku & Duah [37], emphasize that rewards are a prerequisite for employees to share knowledge proactively. Significantly, the moderating role of Digital Mindset (DM) in the KS→KA relationship was also confirmed (β = 0.251, p = 0.000), indicating that when employees have a high digital mindset and active KS, knowledge application efficiency is significantly enhanced. This finding is consistent with the studies of Solberg et al. [14], Kane et al. [19], and Qiao et al. [18], and affirms that DM is an important moderating factor that optimizes the process of converting knowledge into practical action.

## Conclusion and limitations

### Conclusions

Using quantitative research methods through the PLS-SEM model, the results confirm that knowledge-sharing behavior is central to promoting knowledge application abilities at work. Organizational factors such as the reward system,

organizational commitment, strategy, and organizational structure all significantly influence knowledge-sharing behavior, showing the fundamental role of the working environment in building a knowledge culture. Additionally, digital thinking directly affects knowledge application ability and enhances the effectiveness of knowledge transfer in an organization. In particular, when digital thinking is combined with active sharing behavior, the effectiveness of the knowledge application is optimized. These findings strengthen fundamental theories such as social exchange, socio-technical systems, and dynamic capability theories and provide practical implications for designing human resource management and digital transformation policies in Vietnamese commercial banks.

In addition to the theoretical and managerial contributions, the research results also have important implications for the implementation of ESG (Environmental, Social, Governance) goals. First, knowledge sharing and organizational commitment contribute to strengthening the social dimension through increased engagement, cooperation, and collective learning. Organizational strategies oriented towards innovation and knowledge sharing are closely linked to the Environmental dimension when promoting sustainable practices and efficient use of resources. In addition, digital thinking and fair reward mechanisms clearly reflect the Governance dimension, emphasizing transparency, accountability, and responsible technology management. Therefore, developing digital thinking and encouraging knowledge sharing within corporations enhances their innovation capacity and competitive advantage, while also supporting the achievement of ESG goals and contributing to long-term sustainable development.

## Limitations

Despite these significant results, this study had two limitations. First, the survey scope is limited to specific joint-stock commercial banks in Vietnam, especially in Ho Chi Minh City, which reduces its generalizability. Therefore, future studies should expand the sample to include more regions to increase reliability. Second, the study did not include control variables such as age, seniority, education level, or job position, which are factors influencing knowledge sharing or application behavior. Adding control variables in future research would help improve the precision of causal relationship assessment.

The study is applied in the context of commercial banks, where digital transformation is taking place rapidly and comprehensively. In this setting, the demand for a digital mindset among employees has become increasingly critical to meet strategic objectives and enhance operational efficiency in a digital environment. However, there are still certain industries where the adoption of a digital mindset remains limited due to the nature of work, which is often highly manual, rooted in traditional processes, or characterized by a low level of digitalization.

Examples include the jewelry sector, handicraft manufacturing, traditional agriculture, and personal service industries such as hairdressing and spas, fields in which digital transformation primarily focuses on management and marketing rather than daily operational activities. In the future, extending research into these sectors could offer deeper insights into the moderating role of digital mindset across diverse contexts, thereby supporting the development of more tailored training strategies and transformation of roadmaps specific to each industry. Future research could be expanded to include sectors such as jewelry making, traditional crafts, and other labor-intensive industries.

## Supporting information

**S1 Table. Appendix: Measurement items.**
(DOCX)

## Author contributions

**Formal analysis:** Tu Ngoc Tran.

**Investigation:** Tu Ngoc Tran.

**Methodology:** Tu Ngoc Tran.

**Project administration:** Tu Ngoc Tran.

**Resources:** Nguyen Thi Thanh Tam.

**Software:** Khai Minh Tu.

**Supervision:** Tu Ngoc Tran.

**Validation:** Nguyen Thi Thanh Tam, Khai Minh Tu.

**Visualization:** Nguyen Thi Thanh Tam.

**Writing – original draft:** Tu Ngoc Tran, Nguyen Thi Thanh Tam.

**Writing – review & editing:** Tu Ngoc Tran, Khai Minh Tu.

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
