## [Decision Letter · Decision Letter 0]

26 Aug 2025

Dear Dr. Tran,

Thank you for submitting your manuscript to PLOS ONE. After careful consideration, we feel that it has merit but does not fully meet PLOS ONE’s publication criteria as it currently stands. Therefore, we invite you to submit a revised version of the manuscript that addresses the points raised during the review process.

We look forward to receiving your revised manuscript.

Kind regards,

Erhan Atay

Academic Editor

PLOS ONE

Journal Requirements:

2. In the ethics statement in the Methods, you have specified that verbal consent was obtained. Please provide additional details regarding how this consent was documented and witnessed, and state whether this was approved by the IRB

“This research is funded by Saigon University”

4. In the online submission form, you indicated that “The data supporting the findings of this study are available from the corresponding author, T.on request.”

6. Please ensure that you refer to Figure 1 in your text as, if accepted, production will need this reference to link the reader to the figure.

7. Please include a copy of Table 6 which you refer to in your text on page 15.

**Additional Editor Comments:**

Thank you for your continued work on the manuscript titled "[Full Title]". We appreciate the improvements made in the revised version. Your paper presents a valuable contribution to the understanding of digital mindset and knowledge processes, and your model is well-supported by robust quantitative methods.

At this stage, I recommend minor revision prior to acceptance. Below are the key areas to address:

Required Revisions

1. Hypotheses Section:

Ensure that all hypotheses are correctly numbered and listed. Hypothesis 2 appears to be missing, and Hypothesis 4 is listed twice.

2. Consistent Use of Abbreviations:

Once introduced, abbreviations like KS (Knowledge Sharing) and AI (Artificial Intelligence) should be used consistently throughout the text to maintain clarity.

3. Headings and Section Introductions:

Several main and subheadings (e.g., 2., 2.1., 2.3.1.) begin abruptly. Please add short introductory sentences to guide the reader.

4. Literature Review Flow:

Consider merging or streamlining subsections to reduce fragmentation and improve readability.

5. Results Section:

Improve clarity of table descriptions.

Ensure statistical values such as β coefficients are introduced properly, not just during hypothesis confirmation.

Explain the meaning of key findings more directly, rather than reiterating theory.

6. Discussion on Sustainability:

As suggested by Reviewer 2, consider incorporating a brief section connecting your findings to ESG (Environmental, Social, Governance) goals to enhance contextual depth.

7. Data Availability:

PLOS ONE requires that data be publicly available unless legal or ethical restrictions apply. Please update your Data Availability Statement with a justification or ensure the data is deposited in a public repository.

Reviewers' comments:

Reviewer's Responses to Questions

**Comments to the Author**

1. Is the manuscript technically sound, and do the data support the conclusions?

Reviewer #1: Yes

Reviewer #2: Yes

Reviewer #3: Yes

2. Has the statistical analysis been performed appropriately and rigorously?

Reviewer #1: Yes

Reviewer #2: Yes

Reviewer #3: Yes

3. Have the authors made all data underlying the findings in their manuscript fully available?

Reviewer #1: No

Reviewer #2: Yes

Reviewer #3: Yes

4. Is the manuscript presented in an intelligible fashion and written in standard English?

Reviewer #1: Yes

Reviewer #2: Yes

Reviewer #3: Yes

Reviewer #1: The paper presents an interesting model, but several important issues need improvement. The abbreviations KS and AI are not used consistently; once introduced, they should be applied throughout the whole text. After each headline (e.g., 2., 2.1., 2.1.1., 2.3., 2.3.1.) there should be a short explanatory introduction. The literature review contains too many separate headings, which makes the text fragmented and hard to follow.

The hypotheses are also problematic: Hypothesis 2 is missing, while Hypothesis 4 is written twice. The results section is weak—tables are described incorrectly and the presentation is confusing. More focus is needed on explaining the findings, and less on repeating theory. Statistical significance values (β) are not shown in the results; they appear only when confirming or rejecting hypotheses, without explanation of where they were obtained.

Overall, the model is good, but the text requires clearer structure, consistent terminology, and more accurate reporting of results.

Reviewer #2: Dear Author,

Please consider the Sustainability concept regarding the research aim. In this respect, I'd recommend to make a separate Discussion section where the authors could suggest discussing over three dimensions of ESG-goals achievement (Environmental, Social and Governing) in relation to the Article topic.

Reviewer #3: 1. Intelligible and Standard English: Yes.

The manuscript is written in clear, coherent, and standard academic English, adhering to the conventions expected in a scholarly publication. The language is precise, formal, and appropriate for the target audience of PLOS ONE. The structure follows a standard research article format, including sections such as Abstract, Introduction, Literature Review, Hypothesis Development, Research Model and Methods, Results, and Discussions. Each section is logically organized, with clear transitions between ideas. Technical terms like "knowledge sharing," "knowledge application," and "digital mindset" are consistently defined and used in context, supported by references to established theories (e.g., Social Exchange Theory, Dynamic Capabilities Theory). The writing is free of significant grammatical errors, and the use of academic jargon is appropriate for the field of knowledge management and organizational behavior. For example, the Abstract (Page 9) concisely summarizes the study’s objectives, methods, findings, and implications, demonstrating clarity and adherence to standard English.

2. Statistical Analysis: Appropriately and rigorously performed.

The statistical analysis in the manuscript is conducted appropriately and rigorously, employing Partial Least Squares Structural Equation Modeling (PLS-SEM) using SmartPLS software, which is a robust method for analyzing complex relationships in social science research. The document details several key aspects of the analysis:

Measurement Model Validation (Page 20): The manuscript reports high reliability and validity of the constructs, with Cronbach’s Alpha, Composite Reliability (rho_a and rho_c), and Average Variance Extracted (AVE) exceeding thresholds (0.7 for reliability, 0.5 for AVE), indicating strong convergent validity. Discriminant validity is confirmed via the Heterotrait-Monotrait (HTMT) ratio, with all values below 0.85, ensuring constructs are distinct (Table 2, Page 20).

Structural Model Assessment (Page 21): The use of bootstrapping with 5,000 resample iterations ensures robust estimation of path coefficients, standard errors, and t-values for hypothesis testing. The R-square values (0.691 for Knowledge Sharing and 0.511 for Knowledge Application, Table 3, Page 21) indicate good explanatory power, particularly for Knowledge Sharing. Variance Inflation Factor (VIF) values below 5 (Table 4, Page 21) confirm the absence of multicollinearity, ensuring reliable linear relationships.

Effect Sizes (Page 21): The f-square (f²) values categorize the effect sizes (e.g., KS → KA with f² = 0.658 indicates a large effect), following Cohen’s (2013) thresholds (small: ≥0.02, medium: ≥0.15, large: ≥0.35). This demonstrates a rigorous approach to assessing the practical significance of relationships.

Hypothesis Testing (Page 22): All hypothesized relationships are statistically significant (p < 0.05), with t-statistics and p-values reported (Table 5, Page 22), indicating robust testing of the proposed model.

3. Data Availability: No, data available on request from the corresponding author.

The manuscript does not fully comply with PLOS ONE’s data availability policy, which requires all data underlying the findings to be fully available without restriction from the time of publication, except in rare cases with legal or ethical concerns. The Data Availability Statement (Page 6 and Page 8) explicitly states: “The data supporting the findings of this study are available from the corresponding author, T., on request.” This indicates that the data are not publicly available in a repository or within the manuscript/supporting files, but rather are accessible only upon request, which does not meet the journal’s requirement for unrestricted access. The authors selected “No - some restrictions will apply” in the submission form (Page 5), but they do not provide a specific justification (e.g., legal or ethical concerns) for restricting access, which could weaken compliance with the journal’s data policy. While the ethical statement (Page 4) emphasizes confidentiality and anonymity in data collection, it does not explicitly justify the restriction on public data sharing.

**Do you want your identity to be public for this peer review?** For information about this choice, including consent withdrawal, please see our Privacy Policy

Reviewer #1: **Yes:** Nejc Bernik

Reviewer #2: **Yes:** Sergey Barykin

Reviewer #3: No

---

## [Author Response · Author response to Decision Letter 1]

15 Oct 2025

Dear Sir/Madam,

We have revised our manuscript and re-submitted on online system.

Thank you for your time and your consideration.

Regards,

Ngoc Tu

---

## [Decision Letter · Decision Letter 1]

1 Jan 2026

Dear Dr. Tran,

Thank you for submitting your manuscript to PLOS ONE. After careful consideration, we feel that it has merit but does not fully meet PLOS ONE’s publication criteria as it currently stands. Therefore, we invite you to submit a revised version of the manuscript that addresses the points raised during the review process.

The author needs to improve the language expression of the manuscript, as some sentences appear to be redundant at present.

We look forward to receiving your revised manuscript.

Kind regards,

Chunyu Zhang

Academic Editor

PLOS One

Journal Requirements:

Reviewers' comments:

Reviewer's Responses to Questions

**Comments to the Author**

Reviewer #1: All comments have been addressed

Reviewer #3: All comments have been addressed

2. Is the manuscript technically sound, and do the data support the conclusions?

Reviewer #1: Yes

Reviewer #3: Yes

3. Has the statistical analysis been performed appropriately and rigorously?

Reviewer #1: Yes

Reviewer #3: Yes

4. Have the authors made all data underlying the findings in their manuscript fully available?

Reviewer #1: Yes

Reviewer #3: Yes

5. Is the manuscript presented in an intelligible fashion and written in standard English?

Reviewer #1: Yes

Reviewer #3: Yes

Reviewer #1: The authors have addressed all comments from the first review. Abbreviations are now used consistently, literature review is clearer and better structured. The hypotheses have been corrected and numbered properly. The results section is improved, with accurate tables and explained statistical values.

For future submissions, please include one or two sentences below each headline. Avoid placing two consecutive headlines without explanatory text in between.

Reviewer #3: Reviewer Comment:

The paper is empirically solid, with robust use of a staggered Difference-in-Differences (DID) model and various checks, but it has room for enhancement to strengthen its academic rigor, clarity, and impact. The core explanatory variable "Policy" is a dummy (1 if a city opens facilities in a given year, 0 otherwise), which overlooks nuances like the number, type (e.g., wastewater vs. air quality facilities), scale, or frequency of public access. This binary approach may oversimplify the policy's intensity and heterogeneous effects, potentially leading to measurement error or attenuated results. The limitations section acknowledges this but does not explore it deeply.

Under developed Literature Review and Novelty Framing The introduction highlights contributions (e.g., multi-agent regulation, targeted greenwashing measure), but the literature review is brief and China-centric. It cites relevant studies but does not deeply contrast with international contexts (e.g., EU or US disclosure regulations) or emerging theories like stakeholder salience or signalling theory in greenwashing.

Reliance on Specific Data Sources with Potential Coverage Gaps Greenwashing is measured as the absolute difference between standardized Bloomberg ESG disclosure scores and CNRDS ESG performance scores. While innovative, these databases may have incomplete coverage for smaller or non-heavy-polluting Chinese firms, leading to selection bias in the sample (reduced from 21,000 to 4,416 observations). The sample period (2011–2023) is post-policy heavy, but pre-policy trends might be under-represented.

Limited Exploration of Endogeneity Beyond Standard Checks The paper uses PSM-DID, parallel trends, placebo tests, and fixed effects effectively, but does not employ instrumental variables (IV) or other advanced methods to address potential reverse causality (e.g., firms influencing policy implementation) or omitted variables (e.g., local political cycles).

Narrow Scope of Heterogeneity and Mechanisms The motivation (government/public pressure, costs) and heterogeneity (green investors, analysts, subsidies) analyses are strong but limited. For example, it does not explore firm-level traits like ownership type (state-owned vs. private) or industry pollution intensity, which could moderate effects.

Dense Writing and Long Sentences Some sections (e.g., introduction, hypotheses) have lengthy sentences that reduce readability (e.g., multi-clause explanations of mechanisms). Tables are clear, but figures (e.g., parallel trends) lack detailed captions explaining confidence intervals.

Incomplete Limitations and Future Research The limitations section covers data binary nature, unexamined moderators, and archival method limits well, but it is brief and does not quantify impacts (e.g., how binary measure might bias coefficients downward).

Overall, this is a well-executed article with strong empirical contributions to greenwashing literature. Addressing these revisions could elevate it for top journals like PLOS ONE. Prioritize methodological refinements and clarity for the next revision.

**Do you want your identity to be public for this peer review?** For information about this choice, including consent withdrawal, please see our Privacy Policy

Reviewer #1: **Yes:** Nejc Bernik

Reviewer #3: No

---

## [Author Response · Author response to Decision Letter 2]

26 Jan 2026

Thank you so much Reviewer 1 for the careful reading and constructive feedback. We sincerely appreciate your valuable comments, which have helped us improve the clarity, rigor, and overall quality of the manuscript. We have addressed all of Reviewer 1’s comments point-by-point in the revised version and believe the paper is now substantially strengthened.

---

## [Editor Report · Decision Letter 2]

28 Jan 2026

THE MEDIATING ROLE OF KNOWLEDGE SHARING BEHAVIOR AND THE MODERATING ROLE OF DIGITAL MINDSET: EVIDENCE IN VIETNAM

PONE-D-25-42326R2

Dear Dr. Tran,

We’re pleased to inform you that your manuscript has been judged scientifically suitable for publication and will be formally accepted for publication once it meets all outstanding technical requirements.

Kind regards,

Chunyu Zhang

Academic Editor

PLOS One
---

## [Editor Report · Acceptance letter]

PONE-D-25-42326R2

PLOS One

Dear Dr. Tran,

I'm pleased to inform you that your manuscript has been deemed suitable for publication in PLOS One. Congratulations! Your manuscript is now being handed over to our production team.

Kind regards,

on behalf of

Dr. Chunyu Zhang

Academic Editor

PLOS One